# Investigation into Friction and Wear Characteristics of 316L Stainless-Steel Wire at High Temperature

**DOI:** 10.3390/ma16010213

**Published:** 2022-12-26

**Authors:** Mingji Huang, Yili Fu, Xiaoxi Qiao, Ping Chen

**Affiliations:** 1School of Mechanical Engineering, University of Science and Technology Beijing, Beijing 100083, China; 2Shunde Innovation School, University of Science and Technology Beijing, Foshan 528300, China

**Keywords:** metal rubber, 316L stainless-steel wire, high temperature, friction and wear, wear morphology

## Abstract

The damping performance of metal rubber is highly correlated with the tribological properties of the internal metal wires. In this paper, the friction and wear characteristics of 316L stainless-steel wire are investigated under different temperatures, loads, crossing angles, and working strokes. Results show that the friction coefficient increases from 0.415 to 0.635 and the wear depth increases from 34 μm to 51 μm, with the temperature rising from 20 °C to 400 °C. High temperature will soften metal materials and promote the oxidation of metal. Softened materials can be easily sheared and removed under friction action, resulting in high wear depth. However, when a continuous oxide film with high hardness is formed under higher temperature, the oxide film can work as a wear-resisting layer to prevent further wear of the wire to a certain degree. At the same temperature, the loads, crossing angles, and working strokes change the wear resistance by affecting the surface stress, debris removal efficiency, etc., and high temperature will aggravate this change. The results pave the way for the design and selection of materials for high-temperature metal rubber components.

## 1. Introduction

Metal rubber (MR) is made of metal wires through drawing, casting, compression molding, tempering, and vibration stabilization [1,2]. Compared to conventional damping materials, its special material and structure offer enhanced performance, such as corrosion and heat resistance. Metal rubber relies on the mutual friction of internal metal wires to achieve vibration damping. The failure of metal rubber mainly comes from accumulative wear and fracture of the metal wires. Therefore, the service life of metal rubber is closely influenced by the tribological properties of the internal metal wires [3,4,5].

The friction energy dissipation of the metal wire is considered as one of the indicators for metal rubber vibration damping performance [6,7]. Several studies have investigated the influence of lubrication, surface texture, crossing angles, and other factors on the friction and wear characteristics of metal wires [8,9,10]. Dong et al. [6] divided the process of wear of metal wires in metal rubber into two stages: rapid and steady wear. Higher load leads to faster stabilization but also more severe wear. Wu et al. [10] noted that low sliding velocity can lead to sufficient slip between wires, which improved the damping and load-bearing capacity of metal rubber. Ref. [11] shows that the wear resistance of steel wires for metal rubber increases with increasing wire diameter. Chen et al. [12] explored the influence of crossing angles on tribological properties of stainless-steel wire. It was observed that an increase in crossing angle will increase the surface stress and accelerate material spalling, which will reduce the wear resistance of the metal wire. 

Compared to rubber, the damping performance of metal rubber has been confirmed to have better stability at high and low temperatures [13,14,15]. Therefore, the damping performance of metal rubber at high temperature has attracted various researchers [16,17]. Ma et al. [18] found that as the temperature increases, the contact points of metal wires inside the metal rubber increase and the contact status changes. Li et al. [19] further noted that deeper wear scars and more severe adhesion could be observed on metal rubber steel wires at high temperature. However, changes in the wear mechanism of the wires with temperature have not been further analyzed.

Despite considerable research, the influence of high temperature on the friction and wear characteristics of steel wires for metal rubber still has not been studied. Based on this, a quantitative method is employed in this paper to investigate the friction and wear characteristics of 316L stainless-steel wires at high temperature under different conditions. This research aims to provide theoretical support for the selection of materials, enhancing the service life of metal rubber.

## 2. Materials and Methods

Metal rubber is difficult to disassemble and observe after forming, and the environment of a single metal wire is difficult to quantify. In view of the above reasons, this study simplified the contact behavior between metal wires in the process of metal rubber forced vibration as the extruded friction contact between two metal wires. The 316L stainless-steel wires prepared by the cluster drawing process were used in the tests. The steel wires are Ø0.5 mm in diameter and 30 mm in length. Table 1 shows the chemical composition of the tested specimen. 

The tribological performance of stainless-steel wire was tested via metal-wire reciprocating friction testing machine. The testing machine is shown in Figure 1. The lower specimen is mounted on the workbench by the fixture and driven by the motor along with the workbench for linear reciprocating motion. Driven by adjustable motor, the workbench can realize the reciprocating motion of different strokes and speeds. The upper specimen is fixed on the vertical rod by bolt and kept static; adjusting the fixed angle can change the crossing angle of the upper and lower specimens, and the contact loads are provided by the weight placed on the vertical rod. In the experiment, the lower and upper specimens are both 316L stainless-steel wire. In the process of reciprocating movement, the friction force is obtained by measuring the deformation of horizontal beam through a strain gauge. The changes in wear depth drive the vertical displacement of the horizontal beam, then the wear depth can be collected by the displacement sensor at the end of the horizontal beam.

Different temperatures, loads, crossing angles, and strokes on the friction and wear characteristics of 316L stainless-steel wire were studied. The test time was 1200 s, the load range was set to be 10~20 N, the linear reciprocating stroke was 0.4~0.8 mm, the crossing angle was 30~90°, the sliding speed was 240 mm/min, and the test temperature range was 20~400 °C. Table 2 shows the specific test variables. For accuracy of the results, each group should be replicated three times.

## 3. Results

### 3.1. Friction and Wear Characteristics Analysis under Different Temperatures

#### 3.1.1. Friction Coefficient and Wear Depth Analysis under Different Temperatures

Figure 2 shows the friction coefficient and wear depth at different temperatures in group 1 as a function of time. It indicates that there are two stages: unstable wear stage and stable wear stage. In the unstable wear stage (about the first 400 s), there was a sharp increase in the friction coefficient, reaching a peak and then decreasing. With increasing temperature, the peak value of the friction coefficient increases in this stage, rising from 0.451 at 20 °C to 0.769 at 400 °C, and the peak time gradually advances. The wear depth also increases rapidly in the unstable wear stage, and its rise rate increases obviously with increasing temperature.

After entering the stable wear stage, for each certain temperature, the friction coefficient still has an upward tendency, but the increasing speed decreases compared with that in the unstable wear stage, and temperature almost has no effect on the increasing speed. In the stable wear stage, the rising rate of wear depth is lower than that in the unstable wear stage under the same temperature, and this difference gradually increases with increasing temperature. As the temperature rises over 300 °C, in the stable wear stage, the rising rate of wear depth is significantly lower than that at 20~200 °C. The average values of the friction coefficient (about the last 200s, as shown in Figure 2a) and wear depth at the stable wear stage were further obtained, as shown in Table 3. According to the results, an increase in both paraments with increasing temperature can be found. Specifically, the friction coefficient increases from 0.415 at 20 °C to 0.635 at 400 °C. The wear depth increases from 34 μm at 20 °C to 51 μm at 400 °C.

The variation rule of the friction coefficient and wear depth is associated with the wear mechanism and surface state of the friction pair [20,21]. For each certain temperature, the metal wires are in point or linear contact at first, with a large contact stress capable of causing significant tearing and plastic deformation of the micro-convex body on the surface. Therefore, the friction coefficient and wear depth rise rapidly in a short time, which is considered as the unstable wear stage. After a period of time, the surface contact area increases and the contact stress decreases, part of the peeling dust is discharged from the friction pair, and the other part is crushed and extruded during the movement and is applied to the contact surface, forming a bright-white third body bed and weakening the adhesion effect [6,22]. At the end of the unstable wear stage, the friction pair surface becomes smoother and, thus, wear becomes slower. Then, the stable wear stage begins, the formation and discharge of debris enter a stage of dynamic balance, the friction coefficients become relatively stable, and the rise in wear depth also becomes slow from the sharp rise in the unstable wear stage. With increasing temperature, the hardness of the steel matrix decreases, and the friction pair material is easier to peel off, which makes the wear severer. Therefore, in the unstable wear stage, the maximum friction coefficient in the unstable wear stage increases with increasing temperature and so does the rising rate of wear depth.

#### 3.1.2. Wear Mechanism Analysis under Different Temperatures

Figure 3 and Figure 4 show the Scanning Electron Microscope (SEM) morphology and Energy Dispersive Spectroscopy (EDS) images of the worn surface of stainless-steel wire in group 1, respectively. Figure 3 shows that the wear morphology of 316L stainless-steel wires is approximately elliptical pits, whose long and short axes are defined as L and W to characterize the worn surfaces. The area of the worn surfaces can be observed to increase with increasing temperature, and the specific values are listed in Table 4.

According to Figure 3, when the temperature was 20 °C, 100 °C, and 200 °C (Figure 3a,c,e), the main wear mechanisms of 316L stainless-steel wire were adhesive wear and abrasive wear. With increasing temperature, adhesive wear gradually became severe, while the grooves caused by abrasive wear gradually changed from short and deep to shallow and long. In the process of wear, there are hard micro-convex bodies and transfer particles, which can produce micro-cutting on the friction pair surfaces. Due to the surface stress, a part of the transfer particles is aggregated into a group and becomes larger, forming short and deep grooves, while for the transfer particles still with small volume, the micro-cutting grooves are shallow and long. However, the larger particles will be softened with increasing temperature, being easily rolled and coated on the friction pair surfaces, so the number of short and deep grooves decreases [23,24]. While fine transfer particles are also affected by material softening, they still play a certain role in cutting due to their large number and small force area. At the same time, the softening of matrix materials caused by a temperature rise leads to easier migration and peeling of surface materials, which aggravates the adhesive wear.

According to EDS analysis, there were obvious oxygen elements enriched in the worn surfaces at the three temperatures of 20 °C, 100 °C, and 200 °C, which proves that oxidation wear occurs during friction. The oxygen content rose slightly with temperature rising (Figure 4a,c,e), but the oxygen contents were still less than iron content below 200 °C. A study reported that the dense and smooth oxides attached to the friction pair surface can reduce wear to a certain extent [25]. However, if the formation speed of the oxide layer is less than its crushing speed, then the oxide layer cannot protect the matrix material, and it cannot play a good wear-reduction effect. Therefore, in a range of 20~200 °C, although oxide can reduce wear to a certain extent, the negative effect of high temperature, such as material softening, is more obvious. Therefore, the wear depth in the unstable wear and stable wear stages increased with the increase in temperature.

When the temperatures were 300 °C and 400 °C (Figure 4g,i), the worn surfaces displayed more obvious oxygen enrichment, and the oxygen contents were more than that of the iron element. On the whole, the content ratio of oxygen to iron increases with temperature rising for all the five temperatures, which means the oxidation wear becomes more severe at high temperature. Compared with 100 °C and 200 °C, more long and shallow grooves on the worn surface at 300 °C could be observed, and a lot of materials were rolled and coated to cover the grooves after peeling (Figure 3g,h), which proved that abrasive wear and adhesive were more severe. This is because with the increase in temperature, the object of abrasive wear gradually changes from steel matrix to oxide film. The hardness of the oxide film is different from the matrix material. When the temperature rises to 400 °C, the grooves basically disappear, and the worn surface is composed of a lot of material peeling, coating, and relatively smooth oxides, proving that the wear mechanism at this temperature is mainly adhesive wear and oxidation wear [26]. With increasing temperature, the matrix material will further soften, leading to a reduction in wear resistance. On the other hand, the formation rate of oxide will increase obviously as the temperature rises and the formed oxide film will reduce the wear rate. Therefore, in the unstable wear stage, which the oxide film has not formed as a wear-reducing layer, the rising rate of wear depth at 300 °C and 400 °C is significantly higher than that at 20~200 °C (Figure 3b), which proves that the wear behavior is more severe. However, the increase in oxidation layer formation rate at 300 °C and 400 °C leads to a shorter duration of the unstable wear stage than that of 20~200 °C. After entering the stable wear stage, the oxide film covers the worn surface, resulting in a significant decrease in the rising rate of wear depth compared with that of 20~200 °C (Figure 3b).

### 3.2. Friction and Wear Characteristics Analysis under Different Test Paraments

#### 3.2.1. Friction Coefficient and Wear Depth Analysis under Different Test Parameters

As for the metal rubber, its internal metal wires are intertwined and connected, and their contact conditions are not completely uniform. Meanwhile, considerable research has shown that the work conditions have an significant impact on the tribological properties of metal [27,28,29]. Based on this, this section explores the influence rules of loads, crossing angles, and strokes on the friction and wear characteristics for stainless-steel wire at different temperatures. The variation in friction coefficient and wear depth in groups 2, 3, and 4 is shown in Figure 5. It can be seen that for all the different loads, crossing angles, and strokes, the friction coefficient and wear depth increase with increasing temperature.

According to Figure 5a,b, at the same temperature, the friction coefficient decreases and then increases as the load rises, while the wear depth increases monotonically. For different crossing angles (Figure 5c,d), at the same temperature, with increasing crossing angle, both the friction coefficient and wear depth gradually increase. For different strokes (Figure 5e,f), the friction coefficient and wear depth almost increase firstly and then decrease slightly with the decrease in strokes at the same temperature, except for friction coefficients at 20 °C and 100 °C.

#### 3.2.2. Wear Mechanism Analysis under Different Test Parameters

(a)Influence of Loads on Wear Mechanism

Different loads will result in different contact areas and surface stresses for the same stainless-steel wire. When the load rises from 10 N to 15 N, the contact area has a relatively great increment, which reduces the surface stress and further makes the friction coefficient decrease. However, as the load rises from 15 N to 20 N, the increment in the contact area decreases, resulting in an increase in surface stress and friction coefficient (Figure 5a). According to Figure 5b, most of the increments in wear depth from 10 N to 15 N are larger than those from 15 N to 20 N; larger wear depth corresponds to larger contact area, also indicating a surface stress change rule with load.

The SEM images of worn surface under 10 N to 20 N at 20 °C, 200 °C, and 400 °C in group 2 are shown in Figure 6. When the temperature is 20 °C, with the load rising, there are more compacted areas and less grooves in the surface area (Figure 6a,d,h), proving that the severity of adhesive wear increases and abrasive wear decreases with increasing load.

The change tendency of the wear mechanism of steel wire with temperature under 15 N and 20 N is roughly the same as that under 10 N, which means an increase in adhesion wear and oxidation wear. Meanwhile, higher load significantly aggravates the material spalling and plastic deformation for the same temperature. The obvious spalling and the plastic deformation prove that more serious adhesion wear and oxidation wear happened. When the temperature rises to 400 °C, a large area of plastic deformation exists on the worn surface under the loads of 15 N and 20 N (Figure 6c,f,j). It can be known that at higher temperature, higher load will accelerate the shedding of the oxide film and matrix material; therefore, the wear depth increases with increasing load and temperature, respectively.

(b)Influences of Crossing Angles on Wear Mechanism

The worn surface with crossing angles of 30~90° at 20 °C, 200 °C, and 400 °C in group 3 are shown in Figure 7. The shape of the worn surface of stainless-steel wire is different at different crossing angles. As the crossing angles increase, the shape of the worn surface changes from a slender oval to an approximate regular circle, which makes the area of the worn surface decrease as the crossing angle increases [30]. Taking a temperature of 200 °C as an example (Figure 7b,e,h), the worn surface area decreases from 0.12 at 30° to 0.11 at 60° and then to 0.07 at 90°. From Figure 5d, the increase in wear depth as the crossing angle rises from 30° to 60° is relatively small, because the wear area has relatively small changes and so does the surface stress. When the crossing angle rises from 60° to 90°, the area of the worn surface decreases significantly, resulting in a significant increase in surface stress and a rise in wear depth. In summary, the increasing crossing angle results in an increase in contact stress under constant external load, which will lead to an increase in friction coefficient and wear depth.

There is a similarity in the wear mechanism under 30° and 90°. When the temperature is 20 °C, due to the difference in surface stress, the wear trace morphology at 90° is rougher than that at 30°. When the temperature reaches 200 °C, obvious spalling exists on all worn surfaces, but plastic deformation is severe at 90 °C (Figure 7a,b,g,h). When the temperature rises to 400 °C, the influence of different contact stress is more obvious due to material softening. When the parameters are 400 °C and 30°, there is still a relatively smooth wear-reducing layer, while at 400 °C and 90°, there is large plastic deformation and the wear is much more severe. This is because when two steel wires contact each other, the maximum stress appears below the contact surface, while with increasing surface stress, the range and size of the maximum stress also increase [12]. Therefore, under reciprocating friction, the larger crossing angle makes it easier to shed off the material on the contact surface and increase the wear depth. 

(c)Influence of Strokes on Wear Mechanism

SEM images of the worn surface at 20 °C, 200 °C, and 400 °C in group 4 are shown in Figure 8. The area of the worn surface is different under different strokes, but the shape is similar. It can be observed that when the stroke is 0.4 mm, the central part of the worn surface at different temperatures is mainly a compacted area, spalling, and part of plastic deformation. White oxide spalling zones appear at both ends of the worn surfaces (Figure 8a–c), which cannot be observed at higher strokes. This is due to the fact that for smaller strokes, load reversals are also more frequent, which causes surface materials and oxides to face fatigue spalling more easily [31], leading to an increase in wear (Figure 5f). With the increase in stroke, on the one hand, the removal efficiency of debris will decrease, accelerating wear to a certain extent [32], so the wear depth at 0.6 mm is slightly higher than 0.4 mm. On the other hand, the change in stroke will significantly change the area of the worn surface. Under the same external load, a higher stroke will also lead to a larger area of worn surface, thus, reducing the surface stress. Therefore, when the stroke is 0.8 mm, the friction coefficient and wear depth are the lowest.

At temperatures of 20 °C and 200 °C, with the increase in stroke, grooves on the worn surface increase (Figure 8d,e,g,h). This is because under higher stroke, the particles and spalling material under reciprocating friction cannot be easily discharged; sliding extrusion makes it stay on the surface, so there are more grooves on the worn surface. The interference of more particles between friction pairs also leads to an increase in the friction coefficient and wear depth. However, the area of the worn surface will increase and the surface stress will decrease with reciprocating stroke increases. Therefore, it can be observed that the wear decreases when the stroke increases to 0.8 mm. As the temperature rises to 400 °C (Figure 8f,i), the high load reversal frequency and high surface stress in lower stroke make the material spalling more obvious under the softening effect of the material, so the wear depth is higher under lower stroke.

## 4. Conclusions

In this paper, a quantitative method is adopted to investigate the friction and wear characteristics of 316L stainless-steel wire for metal rubber at high temperature. Meanwhile, the factors of load, crossing angle, and stroke parameters are also considered under different temperatures. The salient findings are as follows:(1)The friction coefficient and wear depth of 316L stainless-steel wires increase with increasing temperature, and the friction process can be divided into unstable wear stage and stable wear stage. The temperature increase will accelerate the arrival of the stable wear stage.(2)The wear mechanisms of 316L stainless-steel wire in a range of 20 °C to 400 °C are oxidation wear, abrasive wear, and adhesive wear. When below 200 °C, the proportion of adhesive wear increases while abrasive wear decreases with increasing temperature; the oxidation wear is not the main influence rule. When the temperature rises to more than 200 °C, oxidation wear dominates and the oxide film produced can play a better role in reducing wear.(3)Load, crossing angle, and stoke also affect the tribological behaviors significantly. The effect of load is closely related to surface stress, thus, resulting in nonmonotonic friction coefficient with increasing load. Different crossing angles will obviously change the shape and area of the worn surfaces, leading to different surface stresses and friction coefficients. When the crossing angle reaches 90°, the area of the worn surfaces is the smallest and the wear is the highest. Higher stroke will reduce both surface stress and debris removal efficiency, helping to reduce wear. Constant other factors, lower loads, smaller crossing angle, and higher stroke are more likely to improve the wear resistance.

## Figures and Tables

**Figure 1 materials-16-00213-f001:**
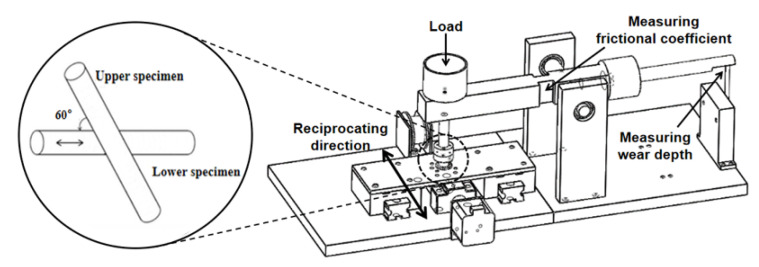
Testing machine.

**Figure 2 materials-16-00213-f002:**
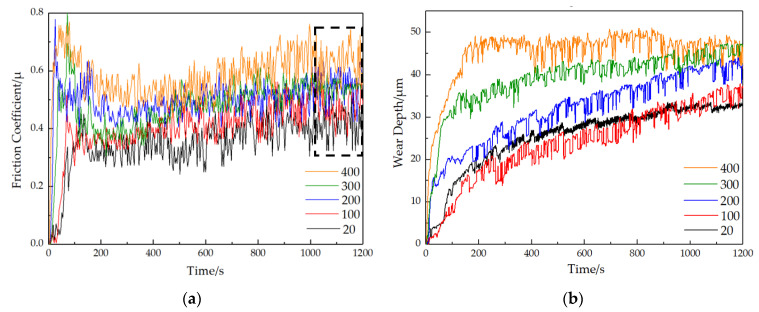
Friction coefficient and wear depth of stainless-steel wire at different temperatures vary with time in group 1: (**a**) friction coefficient; (**b**) wear depth.

**Figure 3 materials-16-00213-f003:**
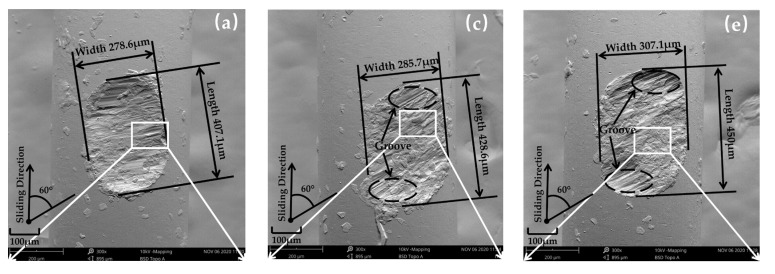
SEM morphology of worn surface of 20~400 °C in group 1: (**a**,**b**) 20 °C; (**c**,**d**) 100 °C; (**e**,**f**) 200 °C; (**g**,**h**) 300 °C; (**i,j**) 400 °C.

**Figure 4 materials-16-00213-f004:**
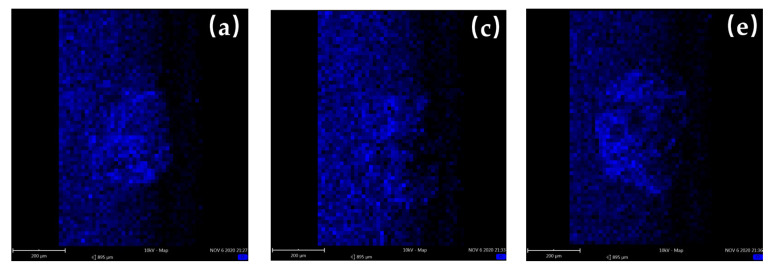
EDS of worn surface and EDS of oxygen distribution of 20~400 °C in group 1: (**a**,**b**) 20 °C; (**c**,**d**) 100 °C; (**e**,**f**) 200 °C; (**g**,**h**) 300 °C; (**i**,**j**) 400 °C.

**Figure 5 materials-16-00213-f005:**
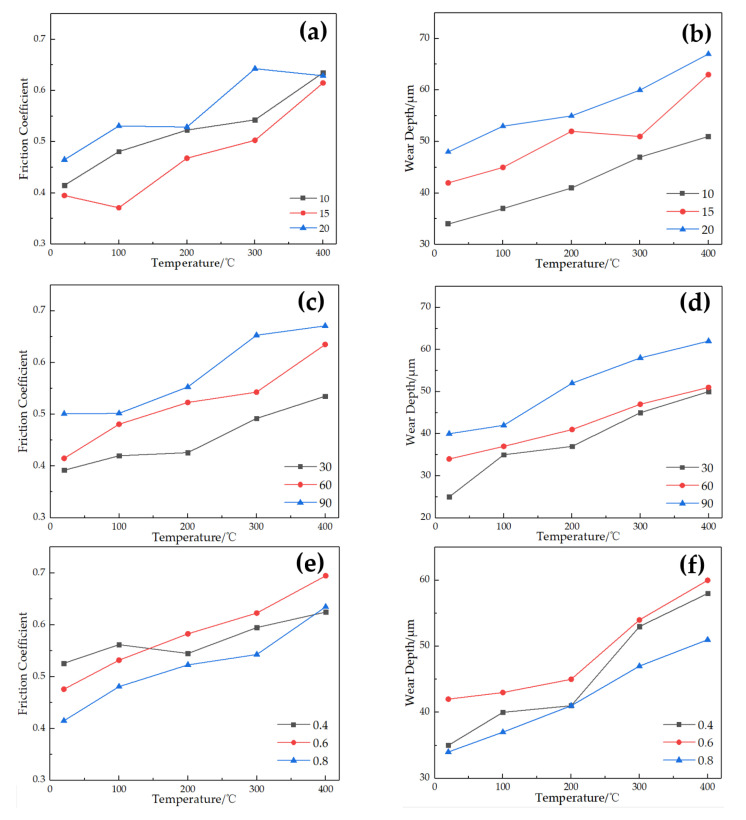
Variation in friction coefficient and wear depth with temperature under different parameters: (**a**,**b**) loads; (**c**,**d**) crossing angles; (**e**,**f**) strokes.

**Figure 6 materials-16-00213-f006:**
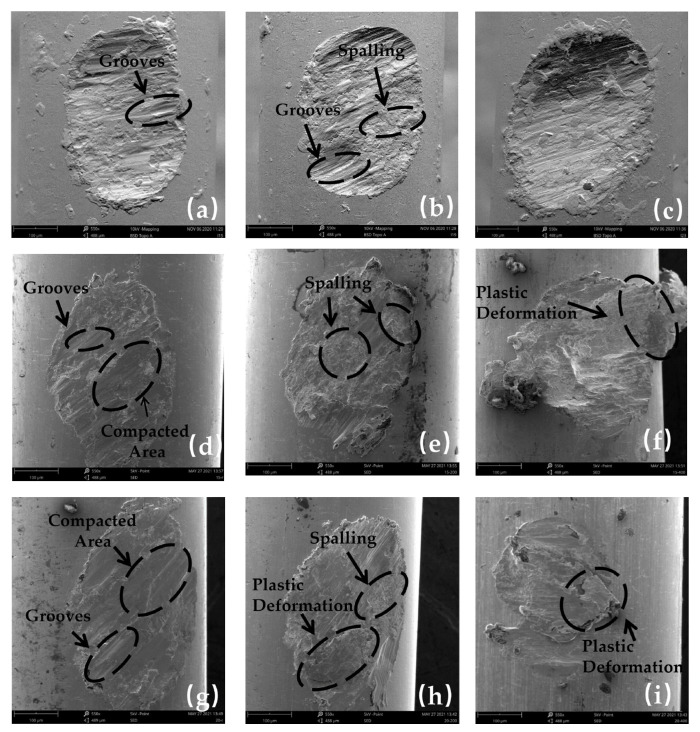
SEM images of worn surface under different loads and temperatures: (**a**) 20 °C—10 N; (**b**) 200 °C—10 N; (**c**) 400 °C—10 N; (**d**) 20 °C—15 N; (**e**) 200 °C—15 N; (**f**) 400 °C—15 N; (**g**) 20 °C—20 N; (**h**) 200 °C—20 N; (**i**) 400 °C—20 N.

**Figure 7 materials-16-00213-f007:**
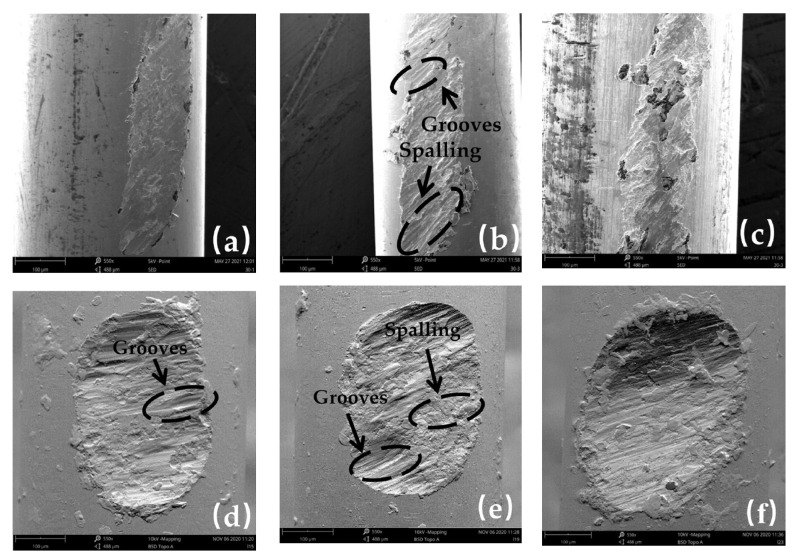
SEM images of worn surface under different crossing angles and temperatures: (**a**) 20 °C—30°; (**b**) 200 °C—30°; (**c**) 400 °C—30°; (**d**) 20 °C—60°; (**e**) 200 °C—60°; (**f**) 400 °C—60°; (**g**) 20 °C—90°; (**h**) 200 °C—90°; (**i**) 400 °C—90°.

**Figure 8 materials-16-00213-f008:**
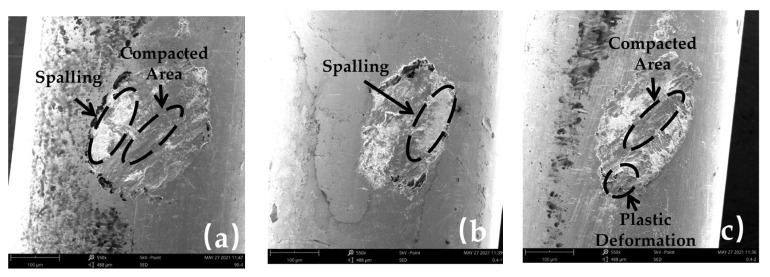
SEM images of worn surface under different strokes and temperatures: (**a**) 20 °C—0.4 mm; (**b**) 200 °C—0.4 mm; (**c**) 400 °C—0.4 mm; (**d**) 20 °C—0.6 mm; (**e**) 200 °C—0.6 mm; (**f**) 400 °C—0.6 mm; (**g**) 20 °C—0.8 mm; (**h**) 200 °C—0.8 mm; (**i**) 400 °C—0.8 mm.

**Table 1 materials-16-00213-t001:** Chemical composition of 316L stainless steel.

C (wt.%)	Si (wt.%)	Mn (wt.%)	P (wt.%)	Cr (wt.%)	Ni (wt.%)	Mo (wt.%)
0.02	0.49	1.03	0.03	17.26	12.0	2.05

**Table 2 materials-16-00213-t002:** Experimental parameters used in different groups.

Test Variables	Group 1	Group 2	Group 3	Group 4
Temperature (°C)	20; 100; 200; 300; 400	20; 100; 200; 300; 400	20; 100; 200; 300; 400	20; 100; 200; 300; 400
Sliding velocity (mm/min)	240	240	240	240
Load (N)	10	10;15;20	10	10
Crossing angle (°)	60	60	30;60;90	60
Stroke (mm)	0.8	0.8	0.8	0.4; 0.6; 0.8

**Table 3 materials-16-00213-t003:** Friction coefficient and wear depth of 316L stainless steel in group 1.

Temperature/°C	20	100	200	300	400
Friction Coefficient	0.415	0.481	0.523	0.543	0.635
Wear Depth	34 μm	37 μm	41 μm	47 μm	51 μm

**Table 4 materials-16-00213-t004:** Friction coefficient and wear depth of 316L stainless steel in group 1.

Temperature/°C	20	100	200	300	400
Width/μm	0.415	0.481	0.523	0.543	0.635
Length/μm	34 μm	37 μm	41 μm	47 μm	51 μm
Area/μm^2^	89,078.3	96,172.8	108,538.1	112,835.4	146,234.8

## Data Availability

The processed data required to reproduce these findings cannot be shared at this time because the data also form part of ongoing research.

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
