# Peer review of "Investigation into Friction and Wear Characteristics of 316L Stainless-Steel Wire at High Temperature"

_materials, 2022, doi:10.3390/ma16010213_

Round 1
Reviewer 1 Report
I reviewed the article "Investigation on Friction and Wear Properties of 316L Stainless Steel Wire in High Temperature". The article is scientifically well organized. Tribological properties have been comprehensively discussed with the findings obtained. The subject of the article falls within the scope of the "Materials" journal. However, the article will be ready for publication after a minor revision. Comments are listed below.
1. A sentence about numerical results can be added to the abstract.
2. In the introduction, the usage areas of metal rubber steel wires are not mentioned. In addition, more up-to-date references on the subject should be given in the introduction.
3. The material ratios in Table 1 should be in the form of "C (wt.%), Si (wt.%), ….
4. Table 2 on page 3 (line 101) should be changed to “table 3”.
5. In Figure 2a, the unit of the horizontal axis title should be added (Friction of coefficicent/µ).
6. On page 4, (line 135), "table 3" should be changed to "table 4".
7. Where is Figure 1? All figure tags and citations in the text should be checked.
8. The article contains numerous typographic and language errors. It should be corrected.
9. The article should be rearranged by taking into account the journal writing rules and citation rules.
10. The article is well organized, but there is a reference problem. First, your reference list contains only one article from the "Materials" journal. If your work is appropriate for the context of this journal, there are many references from this journal. Second, the cited sources should be primary sources. So the indexed area shows the strength of a paper and directly the reliability of your paper. Please make arrangements in this direction.
The Turnitin similarity rate is 23%.

Reviewer 2 Report
The results presented in this study have both a certain scientific novelty and significance, as well as a certain practical utility. The influence of the studied factors: temperature, load, intersection angle, friction time on the friction coefficient and the wear of the steel samples is examined in detail. However, I notice certain minor technical errors, omissions and inaccuracies that, in my opinion, the authors should correct:
- The introduction is not fully convincing, motivated and unambiguous. The purpose of the work is too general - it would be good to specify it.
- Fig. 1 is missing in the manuscript presented to me, and therefore the description of the friction scheme is not clear - the idea of the conditions and scheme of the tribological tests is lost. Also, it is unclear whether the second wire is of the same material (line 63).
- The textual descriptions and analysis of the results of Figs. 3, 4 and 5 are elaborately described, but in too heavy a style, and are difficult to understand. The dependencies and regularities obtained by the authors are not sufficiently clearly and categorically described and formulated, but such are present. I recommend that they be revised to make them accessible to a wider range of readers.
- The text contains undefined and imprecise terms such as "friction properties, wear properties, ".... Wear Properties under Different Temperatures… 83' which should be 'friction and wear characteristics"; "... Its diameter is Ø0.5 mm and its length is 60 30 mm... How long is the length? -The expression is unclear' ; The average values of friction - row 99 - coefficient and wear depth at the stable wear stage were further obtained, as shown - line100- in Table 2." This is not Table 2, but Table 3; The title of Table 4 does not correspond to the content and is the same as that of Table 3; "...; At 200º C as an example of..." line 247 - it is appropriate to add at a temperature of 200º C, etc....
- The conclusions do not reflect and do not fully summarize the obtained dependencies. It would be good if a general comment was made on the mutual influence of the investigated factors on the wear and to indicate those values of the factors at which the wear and the coefficient of friction take minimum values.
